# Primary Production in the Kara, Laptev, and East Siberian Seas

**DOI:** 10.3390/microorganisms11081886

**Published:** 2023-07-26

**Authors:** Soohyun Kim, Kwanwoo Kim, Naeun Jo, Hyo-Keun Jang, So-Hyun Ahn, Janghan Lee, Howon Lee, Sanghoon Park, Dabin Lee, Dean A. Stockwell, Terry E. Whitledge, Sang-Heon Lee

**Affiliations:** 1Department of Oceanography and Marine Research Institute, Pusan National University, Geumjeong-gu, Busan 46241, Republic of Korea; soohyunk1122@gmail.com (S.K.); goanwoo7@pusan.ac.kr (K.K.); jhk7947@naver.com (H.-K.J.); mossinp@pusan.ac.kr (S.P.); ldb1370@pusan.ac.kr (D.L.); 2Library of Marine Samples, Korea Institute of Ocean Science and Technology, Geoje 53201, Republic of Korea; 3Department of Ecology and Conservation, National Marine Biodiversity Institute of Korea, Seocheon 33662, Republic of Korea; naeunjo@mabik.re.kr; 4University of Maryland Center for Environmental Science, Horn Point Laboratory, Cambridge, MD 21613, USA; sahn@umces.edu; 5Départment de Biologie, Université Laval, Québec, QC G1V 0A6, Canada; janghanlee500@gmail.com; 6Marine Ecosystem Research Center, Korea Institute of Ocean Science and Technology, Busan 49111, Republic of Korea; slrkwsx70@gmail.com; 7Institute of Marine Science, University of Alaska Fairbanks, Fairbanks, AK 99775, USA; dastockwell@alaska.edu (D.A.S.); tewhitledge@alaska.edu (T.E.W.)

**Keywords:** primary production, phytoplankton, Kara Sea, Laptev Sea, East Siberian Sea

## Abstract

Understanding of the primary production of phytoplankton in the Kara Sea (KS), the Laptev Sea (LS), and the East Siberian Sea (ESS) remains limited, despite the recognized importance of phytoplankton in the Arctic Ocean. To address this knowledge gap, we conducted three NABOS (Nansen and Amundsen Basins Observational System) expeditions in 2013, 2015, and 2018 to measure in situ primary production rates using a ^13^C-^15^N dual-tracer method and examine their major controlling factors. The main goals in this study were to investigate regional heterogeneity in primary production and derive its contemporary ranges in the KS, LS, and ESS. The daily primary production rates in this study (99 ± 62, 100 ± 77, and 56 ± 35 mg C m^−2^ d^−1^ in the KS, LS, and ESS, respectively) are rather different from the values previously reported in each sea mainly because of spatial and regional differences. Among the three seas, a significantly lower primary production rate was observed in the ESS in comparison to those in the KS and LS. This is likely mainly because of regional differences in freshwater content based on the noticeable relationship (Spearman, *r_s_* = −0.714, *p* < 0.05) between the freshwater content and the primary production rates observed in this study. The contemporary ranges of the annual primary production based on this and previous studies are 0.96–2.64, 0.72–50.52, and 1.68–16.68 g C m^−2^ in the KS, LS, and ESS, respectively. Further intensive field measurements are warranted to enhance our understanding of marine microorganisms and their community-level responses to the currently changing environmental conditions in these poorly studied regions of the Arctic Ocean.

## 1. Introduction

Phytoplankton, as organisms highly sensitive to environmental changes, play a crucial role in marine ecosystems by providing energy to higher trophic levels [1,2,3,4,5]. The primary production of phytoplankton is influenced by various environmental factors, including nutrient availability, light availability, water stratification, and circulation patterns [6,7,8,9,10]. In the Arctic Ocean, these factors are further impacted by significant environmental changes driven by climate change [11,12], such as decreasing sea ice extent [13,14,15] and increasing freshwater content [16,17,18].

Over the past three decades, there has been a rapid reduction in sea ice concentration [19,20,21]. The melting of sea ice contributes to increasing freshwater input [22,23]. Additionally, freshwater inflow from rivers in the Arctic region plays an important role in the primary production of phytoplankton [17,18,23,24]. The presence of freshwater can lead to water stratification [25,26], which reduces vertical mixing and restricts the upward transport of nutrients to the surface waters [27,28]. 

The Kara Sea (KS), Laptev Sea (LS), and East Siberian Sea (ESS), situated on the widest and shallowest continental shelf in the world, are characterized by significant biogeochemical activity involved in the synthesis and processing of organic matter [29,30,31,32]. These three seas receive approximately half of the total freshwater runoff from the Ob, Yenisei, and Lena rivers into the Arctic Ocean [30,33,34,35,36]. Despite their scientific importance, there have been limited studies focusing on the pelagic phytoplankton in this region [20,29,37]. Therefore, the primary goal of this study was to investigate the temporal and spatial variations in primary production driven by major controlling factors in the KS, LS, and ESS. The secondary objective was to establish the contemporary ranges of primary production in these regions, which are among the least biologically understood areas within the Arctic Ocean. Understanding the dynamics of primary production and its controlling factors in these regions is crucial for assessing the impact of environmental changes on marine ecosystems. This study contributes to our knowledge of the responses of marine microorganisms at the community level to a changing Arctic Ocean.

## 2. Materials and Methods

### 2.1. Data Sources, Study Area, and Samplings

The data used in this study were obtained during three NABOS (Nansen and Amundsen Basins Observational System) expeditions conducted in the KS, LS, and ESS: the 9th, 10th, and 11th cruises of RV “Akademik Tryoshnikov” (25 August–19 September 2013, 28 August–26 September 2015, and 31 August–23 September 2018, respectively). The geographical zones of the KS, LS, and ESS were defined based on the classification by Bhavya et al. (2018) [38], as illustrated in Figure 1. 

During these expeditions, data on physical characteristics, including temperature and salinity, were collected using a Seabird SBE9 and a CTD (conductivity–temperature–depth) instrument equipped with dual temperature (SBE3) and conductivity (SBE4) sensors. Water samples for biological and chemical measurements were collected from six light depths, representing 100%, 50%, 30%, 12%, 5%, and 1% penetration of surface irradiance. We calculated each light depth (*D*) at which photosynthetic-available radiation (PAR) at the surface decreases to 1% according to the Lambert–Beer Equation (1), expressed as: 
(1)
$$D=\frac{-ln(\frac{I_{z}}{I_{0}})}{K}$$

where *I_z_* represents the light intensity at each light depth, *I*_0_ is the light intensity at the surface (100%), and *K* is the light attenuation coefficient, typically obtained from measurements using an underwater PAR sensor. However, since the PAR sensor was not available for this particular study, an alternative calculation for *K* was employed using Secchi measurements. 

We employed a different calculation method using Secchi depth (*SD*) measurements, following the equations provided in [39]:
$$K=\frac{1.7}{SD}$$


While it would have been ideal to have direct measurements using underwater PAR sensors or optical instruments, the use of the Secchi disc method provides a practical and widely accepted approach for estimating euphotic depths and Kd (PAR) in the absence of such equipment [38]. Although we do not have specific data from this study, previous studies in the Arctic Ocean have compared light depths obtained from the Secchi disc method with those obtained from underwater PAR sensors [31]. These comparisons have shown a good agreement between the two methods.

The euphotic depth (EPD) was defined as a depth of 1% of the surface underwater PAR. A depth where the density was 0.05 kg m^−3^ higher than the 10 m value was defined as the mixed layer depth (MLD) [40]. The difference in density between the surface and the bottom depths was calculated as the stratification index of the water column (SI) based on [17]. The concentrations of dissolved inorganic nutrients (phosphate, nitrite + nitrate, ammonium, and silicate) from six water depths (100%, 50%, 30%, 12%, 5%, and 1% of the PAR) were analyzed onboard after water samplings using an automated nutrient analyzer (ALPKEM RFA model 300) during the study periods. 

### 2.2. Freshwater Content

The freshwater content (FWC) was calculated following [41]:
$$\mathrm{FWC}=\int_{Zlim}^{0}\left(1-\frac{S\left(z\right)}{Sref}\right)\mathrm{dz}$$

where *Sref* is the reference salinity, *Zlim* is the depth at which *S* and *Sref* are equal, and *S* is in situ salinity. The reference salinity is *Sref*, which is 34.8, the mean salinity of the Arctic Ocean.

### 2.3. Chlorophyll a (chl-a) Concentration Measurement

Water samples (1 L) were collected from the six light depths (100%, 50%, 30%, 12%, 5%, and 1% of the PAR) for assessing total and size-fractionated chl-*a* concentrations. During the three cruises, water samples (0.3 L) were filtered using 25 mm Whatman glass fiber filters (GF/F) for the total chl-*a* concentration. For the size-fractionated chl-*a* concentration, water samples (0.7 L) were progressively filtered through Nucleopore filters with pore sizes of 20 µm and 5 µm, and Whatman GF/F filters with pore size of 0.7 µm. The filters were immediately stored in the freezer (−20 °C) until the analysis on board. After a 24 h extraction with 90% acetone, all the chl-*a* concentrations were quantified using a pre-calibrated Turner Designs model 10-AU fluorometer.

### 2.4. In-Situ Primary Production Experiments

For the measurements of primary production rates, six light depths (100%, 50%, 30%, 12%, 5%, and 1% of the PAR) were determined at each station using a Secchi disk. In order to adjust the light conditions, polycarbonate incubation bottles (1 L) were covered with high-quality lighting screens (LEE filters, UK) corresponding to each light depth from which water sampling was originally obtained. Heavy-isotope-enriched (98–99%) solutions of labeled carbon (NaH^13^CO_3_) and nitrate (K^15^NO_3_) or ammonium (^15^NH_4_Cl) were added to the samples at concentrations of approximately 0.3 mM (^13^CO_2_), 0.8 μM (^15^NO_3_), and 0.1 μM (^15^NH_4_) [38]. The final concentrations of ^13^C enrichment in the sample bottles were approximately 5–10% of the total inorganic carbon in the ambient water during the observation period. The detail method for determining the total inorganic carbon is available in [42,43]. The incubation bottles filled with water sampling were incubated on deck in large polycarbonate incubators under natural light conditions for 4–6 h. For primary production rates, one third of the incubated samples (0.3 L) were filtered using precombusted (450 °C) GF/F filters (25 mm in diameter), and then the filters were stored in a freezer (−20 °C) until a further analysis. After each field cruise, the filters were fumed with a strong hydro acid in a desiccator overnight to remove carbonate and then dried with a freeze dryer for two hours at a home laboratory. The concentration of particulate organic carbon (POC) and nitrogen (PON) and ^13^C and ^15^N abundances were determined by a Thermo Finnigan Delta + XL mass spectrometer at the stable isotope laboratory of University of Alaska (Fairbanks, AK, USA). Only primary production rates based on carbon uptake rates were used for this study. Dark carbon uptake rates obtained from dark incubation bottles were subtracted from, corresponding to each light carbon uptake rate, in the calculations for all primary productivities [44]. Since the Arctic Ocean has a 24 h photoperiod during the summer [45,46], the daily primary production rates in August and September were calculated by multiplying the hourly primary production rates by 24 h for comparison. 

In this study, the euphotic water-column-integral primary production was derived by integrating the primary production rate at each of the six light depths at a given station vertically using the trapezoidal rule. This method involved dividing the water column into six different light depths and calculating the area under the primary production profile within each depth range. By summing up the areas from all six depths, an approximation of the euphotic water-column-integral primary production was obtained.

## 3. Results

### 3.1. Hydrographic Environmental Conditions

The average temperatures and salinities of the upper 100 m for each cruise are presented in Table 1, Table 2 and Table 3. In 2013 and 2015, the annual average temperature showed a similar pattern across the study areas. The KS exhibited the highest temperature, followed by the LS and the ESS. However, these differences were not statistically significant (Kruskal–Wallis test, *p* > 0.05). In 2018, the average temperature in the LS was slightly lower than that in the ESS, but no significant difference was observed (Mann–Whitney U test, *p* > 0.05). 

Regarding salinity, the annual average salinities in the ESS were significantly lower compared to the KS and LS (Kruskal–Wallis test, *p* < 0.05). The EPD was analyzed to understand vertical light penetration. In the KS, the EPD show a deeper value in 2013 compared to in 2015, but the difference was not statistically significant (Mann–Whitney U test, *p* > 0.05) (Table 1, Table 2 and Table 3). In the LS, the average EPD was deepest in 2013 (average ± S.D. = 50.6 ± 10.3 m) among the three years while, in 2015 and 2018, the EPD were 33.3 ± 5.6 m and 36.7 ± 6.7 m, respectively. In the ESS, the EPD was significantly deeper in 2015 compared to 2013 and 2018 (Kruskal–Wallis test, *p* < 0.05). 

No significant differences in the SI were observed in the KS, LS, and ESS among the three years of observation (Mann–Whitney U test, Kruskal–Wallis test, *p* > 0.05).

### 3.2. Nutrient Concentrations

In the KS, no significant differences were found in the concentrations of ammonium, nitrite + nitrate, phosphate, and silicate between 2013 and 2015 (Mann–Whitney U test, *p* > 0.05) (Figure 2). However, in the LS, the ammonium concentration was significantly higher in 2018 (average ± S.D. = 2.22 ± 1.22 µM) compared to 2013 (average ± S.D. = 0.26 ± 0.14 µM) and 2015 (average ± S.D. = 1.19 ± 1.95 µM) (Kruskal–Wallis test, *p* < 0.05). Additionally, the average phosphate concentration in 2018 (average ± S.D. = 0.18 ± 0.04 µM) was significantly lower than those in 2013 (average ± S.D. = 0.43 ± 0.14 µM) and 2015 (average ± S.D. = 0.24 ± 0.10 µM) (Kruskal–Wallis test, *p* < 0.05). No significant interannual differences were observed for the concentrations of nitrite + nitrate and silicate in the LS. In contrast, in the ESS, all the dissolved inorganic nutrients showed significant differences among the three years observed (Kruskal–Wallis test, *p* < 0.05). In accordance with the definitions provided by [47,48], nutrient limitations can be classified as follows: nitrogen (N) limitation occurs when the ratio of DIN (ammonium + nitrite + nitrate) to phosphate (P) is less than 10 and the ratio of silicate (Si) to DIN is greater than 1. P limitation is observed when the Si to P ratio exceeds 22 and the DIN to P ratio is greater than 22. Si limitation occurs when the Si to P ratio is less than 10 and the Si to DIN ratio is less than 1. Based on these criteria, our results (Figure 3) suggest that phytoplankton at most stations within the ESS could have experienced N limitation during the observation period. Furthermore, our results reveal that approximately 40% of the stations in the LS exhibited Si-limited conditions (Figure 3). However, there were no distinct nutrient-limited conditions observed in the KS during the study period.

### 3.3. Chlorophyll-a Concentrations

The chl-*a* concentrations, which were obtained by integrating measurements from the EFD, are presented in Figure 4 and summarized in Appendix A. In the KS, the statistical analysis using the Mann–Whitney U test did not reveal any significant difference in the chl-*a* concentrations between these two years (*p* > 0.05). Similarly, in the LS, although there were variations in the chl-*a* concentrations across the years, the statistical analysis using the Kruskal–Wallis test did not indicate a significant difference among the three years (*p* > 0.05). In the ESS, the chl-*a* concentrations also exhibited variability. Importantly, the average chl-*a* concentrations in 2015 were significantly higher (*p* < 0.05) compared to those in both 2013 and 2018 based on the statistical analysis (Figure 5). 

In summary, while there were temporal variations in the chl-*a* concentrations within each region, the statistical tests did not identify significant differences in the KS and LS regions. Conversely, in the ESS, the chl-*a* concentrations in 2015 were significantly higher compared to the other two years.

The composition of phytoplankton based on size-fractioned chl-*a* concentrations in this study is illustrated in Figure 6. In the KS region, pico-sized phytoplankton contributed 47.9% (±16.3%) in 2013 and 37.8% (±20.4%) in 2015. Micro-sized phytoplankton accounted for 22.5% (±12.0%) in 2013, while nano-sized phytoplankton contributed 36.5% (±17.7%) during the same period. In 2015, the contribution of micro-sized phytoplankton increased to 29.5% (±11.6%), whereas the contribution of nano-sized phytoplankton decreased to 25.7% (±8.1%).

In the LS region, pico-sized phytoplankton consistently represented a significant fraction across the three years. In 2013, it accounted for 61.3% (±13.9%), which increased to 70.2% (±13.7%) in 2015 and 77.2% (±11.3%) in 2018. The fractions of micro-sized and nano-sized phytoplankton in 2013 were 15.9% (±13.7%) and 22.8% (±7.9%), respectively. In 2015, these fractions decreased to 12.8% (±12.5%) and 17.0% (±2.9%), and in 2018, they further decreased to 9.6% (±8.3%) and 13.3% (±5.0%), respectively.

In the ESS region, the contributions of pico-sized phytoplankton varied among the years. In 2013, they accounted for 73.2% (±2.2%) of the composition, which decreased to 44.6% (±21.3%) in 2015, and then increased again to 59.0% (±24.0%) in 2018. The compositions of micro-sized and nano-sized phytoplankton were 9.1% (±1.5%) and 17.8% (±0.7%) in 2013, 38.8% (±21.1%) and 16.6% (±3.2%) in 2015, and 29.2% (±24.5%) and 11.9% (±1.9%) in 2018, respectively.

In summary, the proportions of phytoplankton varied across the studied regions and years. Pico-sized phytoplankton dominated in the LS region throughout the three years, while the KS region showed variations between pico-sized, nano-sized, and micro-sized phytoplankton. The ESS region exhibited changes in the contributions of pico-sized and micro-sized phytoplankton, with noticeable variations between the years.

### 3.4. Primary Production Rates

The hourly primary production rates integrated from the EPD are presented in Figure 7 and summarized in Appendix A. In the KS, the hourly primary production rates displayed a narrowing range from 2013 to 2015, indicating a more consistent and stable productivity during the latter year. In the LS, there were noticeable fluctuations in the range of hourly primary production rates across the years. The highest range was observed in 2013, indicating a wider variability in productivity during that year. Similarly, in the ESS, the range of hourly primary production rates also varied among the years. The narrowest range was observed in 2018, suggesting a more consistent productivity during that period.

When examining the average hourly primary production rates (Figure 8), a decrease was observed in both the KS and LS regions from 2013 to 2015. This indicates a potential decline in productivity during that period, with the phytoplankton community in these regions producing less carbon on average. In contrast, the average hourly primary production rates in the ESS region increased from 2013 to 2015 before declining in 2018. 

A significant interannual difference was observed only in the ESS region, indicating that the productivity patterns in this region were more variable compared to the KS and LS regions. This could be attributed to the complex interplay of environmental factors, nutrient availability, and the dynamics of the phytoplankton community in the ESS region, leading to distinct variations in primary production rates across the studied years.

### 3.5. Freshwater Content and Freshwater Inventory

The spatial distribution of the FWC for the KS, LS, and ESS during the three NABOS cruises are shown in Figure 9 and summarized in Appendix A. The FWC ranges in the KS were relatively narrow, with no significant differences observed between 2013 and 2015 (Mann–Whitney U test, *p* > 0.05). The average FWC in the KS was around 2.45 m in 2013 and 2.55 m in 2015 (Figure 10). In the LS, the FWC ranges were wider compared to the KS, varying from 1.95 to 6.94 m. However, there were no significant interannual differences observed in the FWC among the three years (Kruskal–Wallis test, *p* > 0.05). The average FWC in the LS was 3.70 m. In the ESS, the FWC exhibited the widest range, ranging from 4.82 to 17.1 m. Similar to the LS, no significant interannual differences were found in the FWC within the ESS (Kruskal–Wallis test, *p* > 0.05). When comparing the FWC across the three regions, a significant regional difference was observed (Figure 10). The FWC in the ESS was substantially higher than those in the KS and LS, indicating a greater freshwater influence in the ESS.

Furthermore, an analysis of the relationship between FWC and primary production rates revealed a significant negative correlation (Spearman, *r_s_* = − 0.714, *p* < 0.05) (Figure 11). This suggests that higher FWC was associated with lower primary production rates of phytoplankton in the studied areas.

## 4. Discussion

### 4.1. Interannual Variations in Primary Production Rates among the Three Seas

During our observation period in 2013, 2015, and 2018, the water-column-integral chl-*a* concentrations exhibited interannual variability but were not significantly different in the KS and LS (Figure 5). In contrast, the interannually averaged chl-*a* concentration in the ESS showed a significant difference (Kruskal–Wallis test, *p* < 0.05) among the three years, with the lowest concentration observed in 2018. While the presence of pronounced subsurface chl-*a* maximum layers is characteristic of the KS during the middle of the summer period [49], no such subsurface chl-*a* maximum layers were found in the KS or the other two regions in this study.

Throughout the observation period, pico-sized phytoplankton appeared to dominate in the KS (average ± S.D. = 43.6 ± 17.4%), the LS (average ± S.D. = 68.3 ± 14.4%), and the ESS (average ± S.D. = 55.1 ± 22.6%), although there was interannual variation in the size compositions of chl-*a* (Figure 6). Small phytoplankton generally contribute a substantial portion of the biomass and primary production of phytoplankton in the Arctic Ocean, except in polynyas and highly productive regions [46,50,51]. 

No significant interannual differences in the primary production rates were observed in the KS and LS, whereas the primary production rates in the ESS were significantly lower in 2018 compared to 2013 and 2015 (Kruskal–Wallis test, *p* < 0.05), similar to the pattern observed for chl-*a* concentration (Figure 8). Based on the positive correlation between the primary production rates and the nitrite + nitrate concentrations (Spearman, *r_s_* = 0.537, *p* < 0.05), the different nutrient conditions, especially nitrite + nitrate availability, could be a major factor driving the interannual differences in the primary production rates in the ESS during the study period. Additionally, variations in phytoplankton biomass may also play a significant role in the interannual differences in primary production rates in the ESS, as evidenced by the strong correlation between primary production rates and chl-*a* concentrations observed in this study (Spearman, *r_s_* = 0.660, *p* < 0.01). It is well known that primary production in the Arctic seas is primarily limited by major inorganic nutrient concentrations, especially nitrate concentrations, toward the end of the growing season [17,23,27]. Previous studies have shown that low nutrient and chl-*a* concentrations can result in low primary production rates in the Chukchi Sea [23,24]. Similarly, [17] reported strong positive relationships between nitrate concentrations and primary production rates and chl-*a* concentrations in the Pacific Arctic Ocean. In the ESS, [52] also found that the water-column-integrated primary production is largely dependent on the dissolved inorganic nitrogen and chl-*a* concentrations during the September period. 

When comparing the regional primary production rates, those in the ESS were significantly lower than in the KS and LS in this study (Figure 8). The regional difference in primary production rates between the ESS and the other two seas could largely be attributed to differences in FWC. The FWC in the ESS was significantly higher than those in the other two seas during the observation period (Figure 10), consistent with previous findings in [53] showing a general increase in the FWC from the KS and LS to the ESS in 2015. In the KS, primary production rates are largely influenced by intensive river discharge, mainly from the OB and Yenisei rivers, during the summer and autumn [54]. However, in this study, large effects from river discharge in the KS were not observed based on freshwater content (Figure 11).

The flow of rivers and sea ice meltwater are the main sources of freshwater in the Arctic Ocean [18,22]. The ESS receives freshwater from rivers running along the Siberian continental slope [33,53,55,56,57]. Indeed, the percentages of river water in the ESS in 2015 and 2018 were significantly higher than those in the KS and LS during the study period. The high FWC in the ESS likely enhanced water column stratification, which, in turn, affected the nutrient reservoir and its replenishment from deep water in the Pacific Arctic Ocean [17]. In this study, the index of the vertical stratification of the water column (i.e., SI) in the ESS (average ± S.D. = 3.94 ± 1.25 kg m^−3^) was significantly higher than those in the KS (average ± S.D. = 1.89 ± 1.72 kg m^−3^) and LS (average ± S.D. = 2.95 ± 1.15 kg m^−3^). Furthermore, a strong positive relationship between FWC and SI was observed (Spearman, *r_s_* = 0.721, *p* < 0.01). According to [27], such increased stratification constrains the vertical nitrate flux and reduces overall nitrate concentrations in the Canada Basin. Indeed, nitrite + nitrate concentrations were notably lower in the ESS compared to the other two seas and showed a negative relationship with SI, except at the station AT071 in 2015 where particularly high concentrations were detected due to its geographical proximity to land. N-limited conditions were observed at most of the stations in the ESS during this study period (Figure 3). Such low nitrite + nitrate conditions, occurring with high FWC and significant stratification, can lead to low chl-*a* concentrations in the Arctic Ocean [17,23,58,59]. We observed that chl-*a* concentrations in the ESS were significantly lower than those in the other seas (Figure 5). Moreover, there was a highly significant positive relationship between primary production rates and chl-*a* concentrations (Spearman, *r_s_* = 0.518, *p* < 0.01). 

Recent hydrographic data provided compelling evidence of a significant increase in freshwater content within the Beaufort Gyre from 2003 to 2018, which is attributed to persistent anticyclonic atmospheric wind forcing, sea ice melt, redirection of Mackenzie River discharge, and the inflow of low-salinity waters originating from the Pacific Ocean [60]. This finding is in contrast to the observed decrease in Siberian river waters that occurred after the 1990s [60]. The authors of [61] further observed the presence of specific phytoplankton, such as *Micromonas*, in the ESS during their observations in 2012 and 2015, suggesting their origin to be the Beaufort Gyre. However, during our observation period, it is unclear which sources contributed to the freshwater content in the ESS. Further investigation is necessary to understand the response of the phytoplankton community to variations in freshwater content within the Beaufort Gyre. 

In summary, the higher FWC stratified the water column, inhibiting nutrient supply from the deep layer and reducing chl-*a* concentrations in the ESS. Consequently, the ESS consistently exhibited lower primary production rates, mainly due to the lower phytoplankton biomass in comparison to the KS and LS. These findings contribute to our understanding of the complex interactions between freshwater content, nutrient availability, and primary production in the Arctic Ocean. 

### 4.2. Comparisons of Primary Production Rates between This and Previous Studies

The regional-averaged daily primary production rates were derived from the hourly primary production measurements conducted during the summer period (Table 4), assuming a 24 h photoperiod per day in the Arctic Ocean [45,46]. Our study found that the average daily primary production rates in the KS and LS, based on the two or three years of observations, were 99 ± 62 and 100 ± 77 mg C m^−2^ d^−1^, respectively. These values were somewhat higher than those reported previously in the KS and LS (Table 1). In contrast, the three year-average daily primary production rate in the ESS fell within the range reported in previous studies conducted from July to September. Although no significant differences were observed in the primary production rates between this and previous studies (Kruskal–Wallis test, *p* > 0.05), it is worth noting that the rates exhibited a wide range within each region. The differences in primary production rates between this and previous studies can be mainly attributed to the spatial and seasonal variations in the KS, LS, and ESS. For example, most of the stations in the LS were located on continental slopes (Figure 1), where an upwelling of nutrient-rich waters at the shelf border is known to enhance primary production [62]. In the ESS, previous productivity measurements were conducted near the northern Chukchi Sea, which experiences relatively higher primary production at the shelf break due to the transport of nutrient-rich Bering Sea water by the shelf break jet and strong easterly winds promoting an upwelling of nutrient-rich water from the deep Canada Basin [37,63,64]. However, the majority of our measurements in this study were conducted farther away from the northern Chukchi Sea (Figure 1).

Regarding the seasonality of primary production, several studies have examined this aspect in the Arctic Ocean, including the KS, LS, and ESS [37,62,65,66]. However, primary production measurements in the KS, LS, and ESS have been scarce and under-sampled to date. Therefore, due to the limited number of field measurements, it is challenging to fully explain the differences in primary production rates between this and previous studies. As a result, we attempted to compile the currently available data from this and previous studies during the phytoplankton growing seasons, which were as follows: 8–220 mg C m^−2^ d^−1^ (average ± S.D. = 49 ± 46 mg C m^−2^ d^−1^), 6–421 mg C m^−2^ d^−1^ (average ± S.D. = 71 ± 58 mg C m^−2^ d^−1^), and 14–139 mg C m^−2^ d^−1^ (average ± S.D. = 51 ± 30 mg C m^−2^ d^−1^) in the KS, LS, and ESS, respectively. Rough estimates of the annual primary production ranges, based on a 120-day growing season in the Arctic Ocean [37,44,45], were 0.96–2.64 g C m^−2^ (average ± S.D. = 5.88 ± 5.52 g C m^−2^) in the KS, 0.72–50.52 g C m^−2^ (average ± S.D. = 8.52 ± 6.96 g C m^−2^) in the LS, and 1.68 –16.68 g C m^−2^ (average ± S.D. = 6.12 ± 3.60 g C m^−2^) in the ESS. 

Although there are limited annual primary production measurements based on field observations in the KS and LS for comparison purposes, a few studies have provided estimates for the ESS. The average annual primary production (6.12 g C m^−2^) in the ESS from our study is relatively consistent with indirect estimates (9.6 g C m^−2^) derived from dissolved inorganic carbon measurements [33] and direct measurements (8 g C m^−2^) based on the ARCSS-PP database [37]. However, satellite-based estimations of primary production (101–121 g C m^−2^) in the KS, LS, and ESS for the period 1998–2009 are considerably higher [20]. Discrepancies between in situ and satellite-based approaches for primary production estimations are common in various oceans, particularly in the Arctic Ocean [67]. The potential reasons for the large discrepancies have been extensively discussed in [32]. 

This study presents valuable insights into the contemporary ranges of primary production rates in the KS, LS, and ESS. The observed spatial and seasonal differences in primary production rates emphasize the need for comprehensive studies covering a wide range of regions and seasons. Furthermore, the validation and improvement of satellite-based estimations are necessary to accurately assess primary production in the Arctic Ocean. Intensive field measurements, combined with investigations of biogeochemical processes and ecosystem dynamics, will contribute to a better understanding of the impacts of environmental changes on marine microorganisms at both species and community levels in the KS, LS, and ESS, considering the influence of different phytoplankton communities on primary production and biogeochemical processes [51].

## Figures and Tables

**Figure 1 microorganisms-11-01886-f001:**
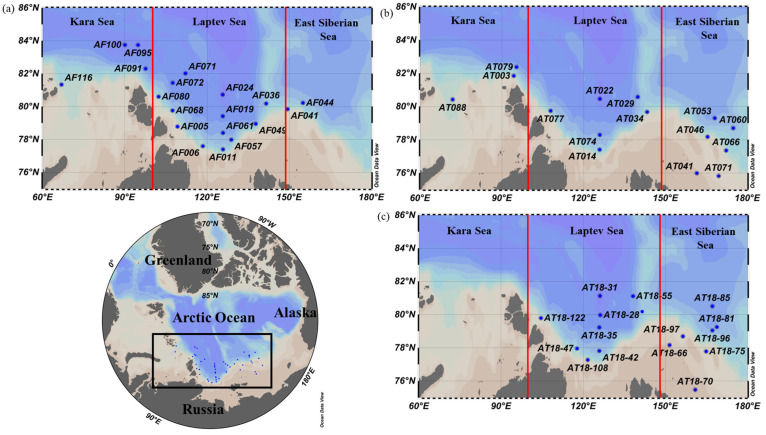
Map of the locations of productivity measurement stations in (**a**) 25 August–19 September 2013, (**b**) 28 August–26 September 2015, (**c**) 31 August–23 September 2018.

**Figure 2 microorganisms-11-01886-f002:**
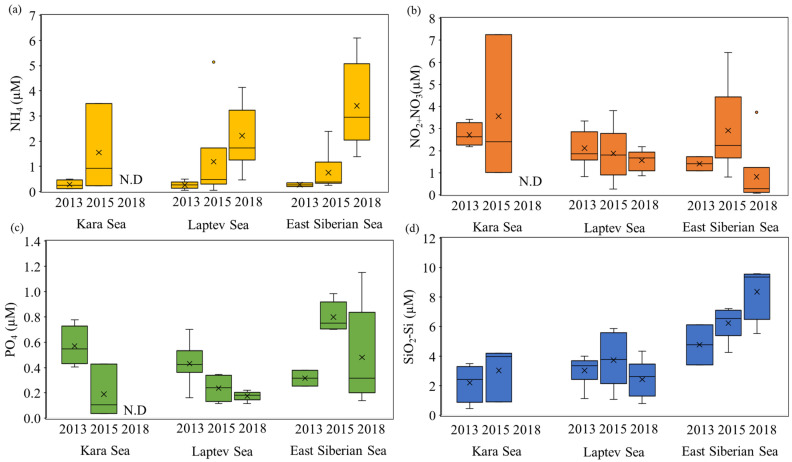
Comparison of dissolved inorganic nutrients in the Kara Sea, Laptev Sea, and East Siberian Sea in 2013, 2015, and 2018. N.D: no data available. (**a**) ammonium; (**b**) nitrite + nitrate; (**c**) phosphate; (**d**) silicate.

**Figure 3 microorganisms-11-01886-f003:**
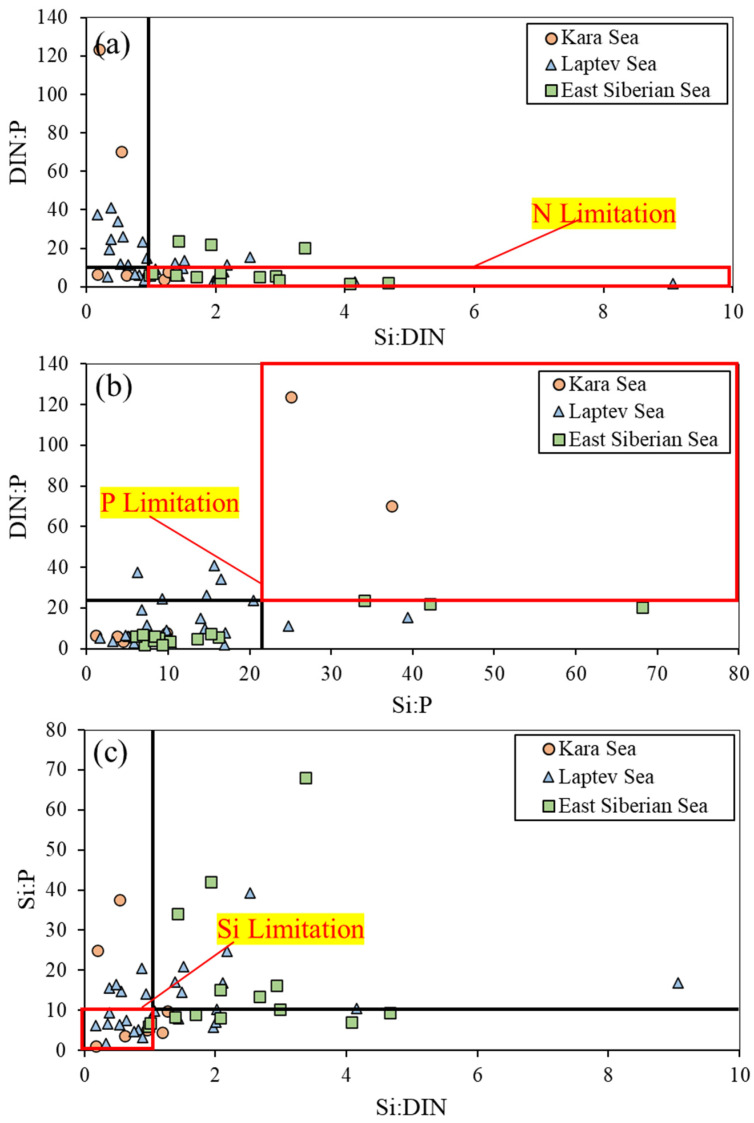
(**a**–**c**) Scatter diagrams of atomic nutrient ratios in all years in the Kara Sea, Laptev Sea, and East Siberian Sea. The red square box represents each nutrient limitation.

**Figure 4 microorganisms-11-01886-f004:**
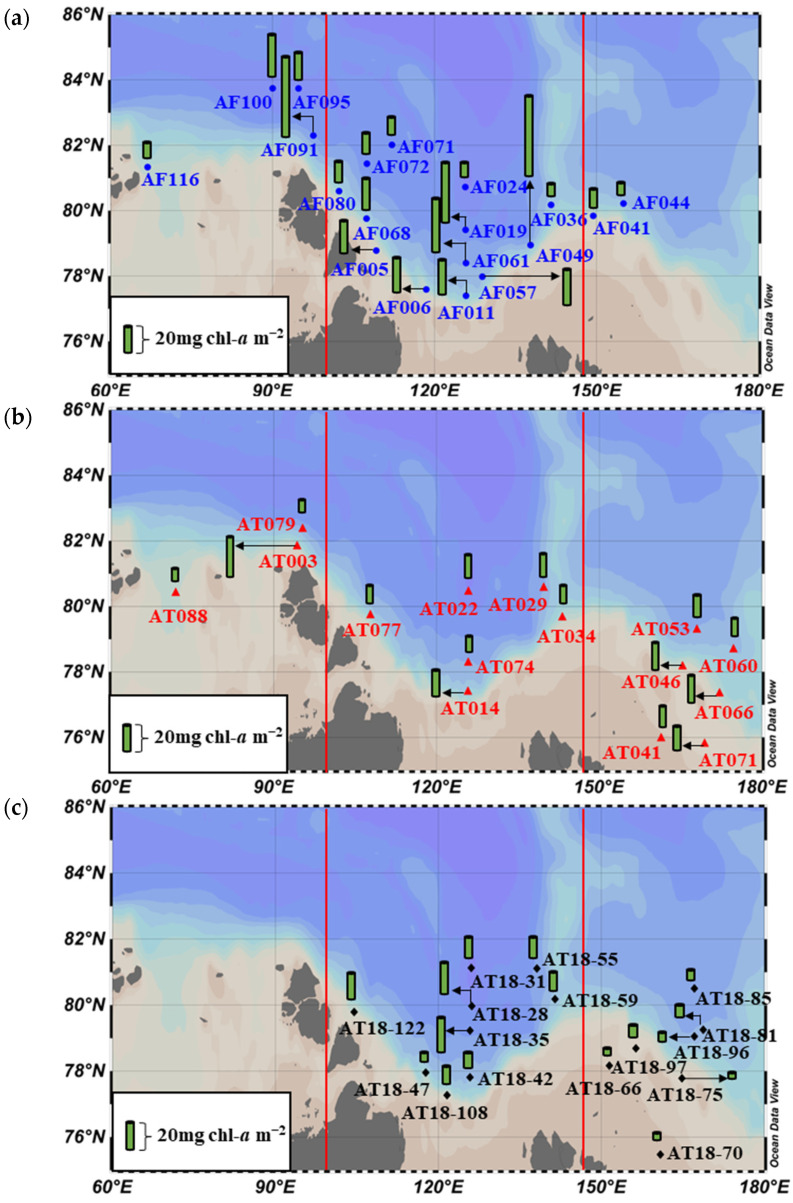
Spatial distribution of the chl-*a* concentrations of phytoplankton during (**a**) 2013, (**b**) 2015, and (**c**) 2018.

**Figure 5 microorganisms-11-01886-f005:**
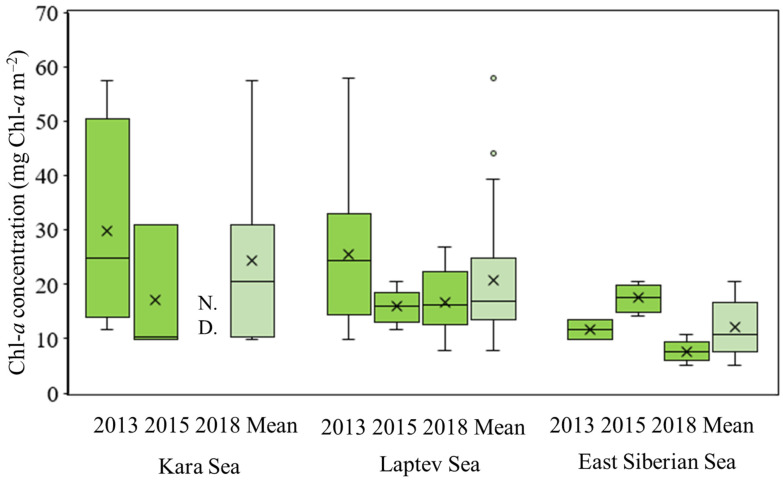
Yearly average chl-*a* concentrations in the Kara Sea, Laptev Sea, and East Siberian Sea in 2013, 2015, and 2018. N.D.: no data available.

**Figure 6 microorganisms-11-01886-f006:**
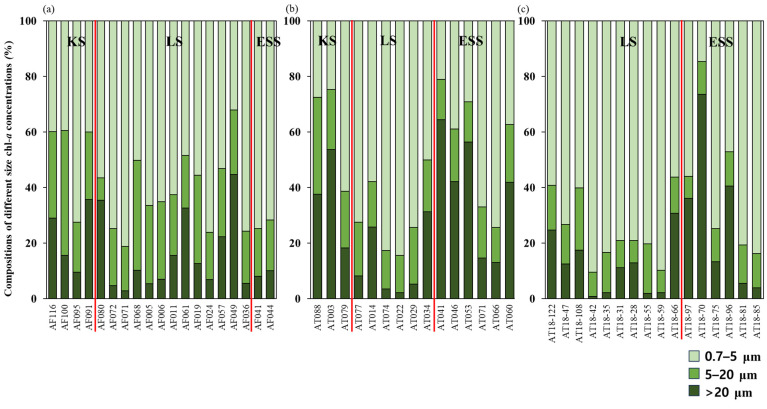
Compositions of different size chl-*a* concentrations at the productivity measurement stations during (**a**) 2013, (**b**) 2015, and (**c**) 2018.

**Figure 7 microorganisms-11-01886-f007:**
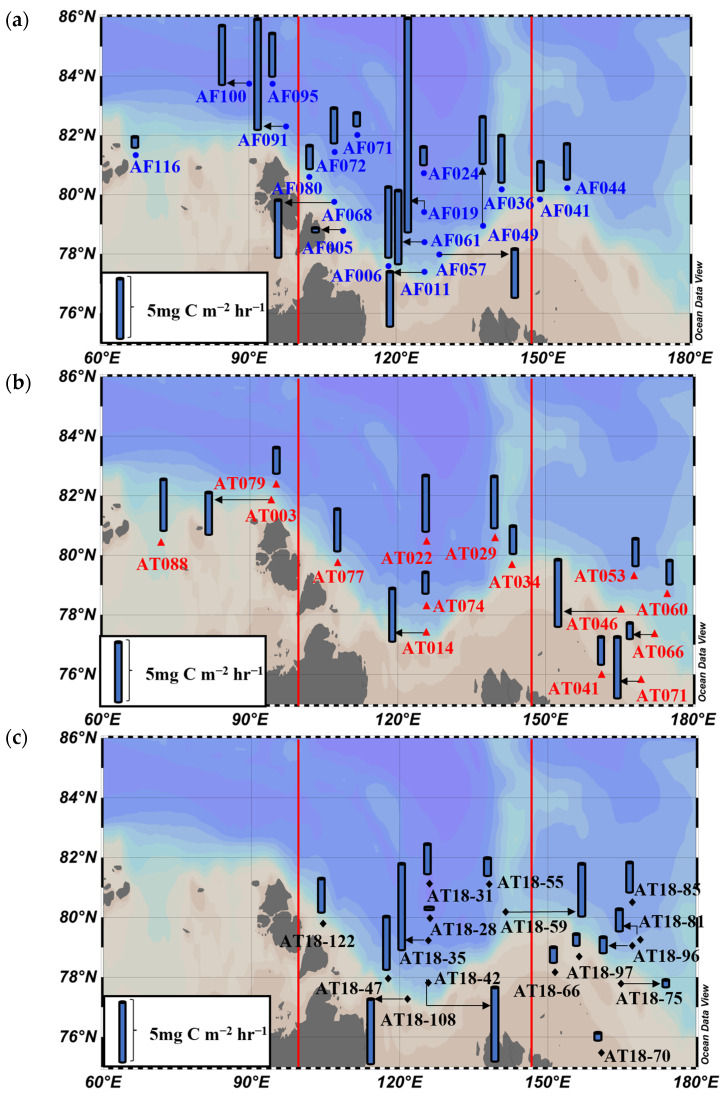
Spatial distribution of hourly primary production rates of phytoplankton during (**a**) 2013, (**b**) 2015, and (**c**) 2018.

**Figure 8 microorganisms-11-01886-f008:**
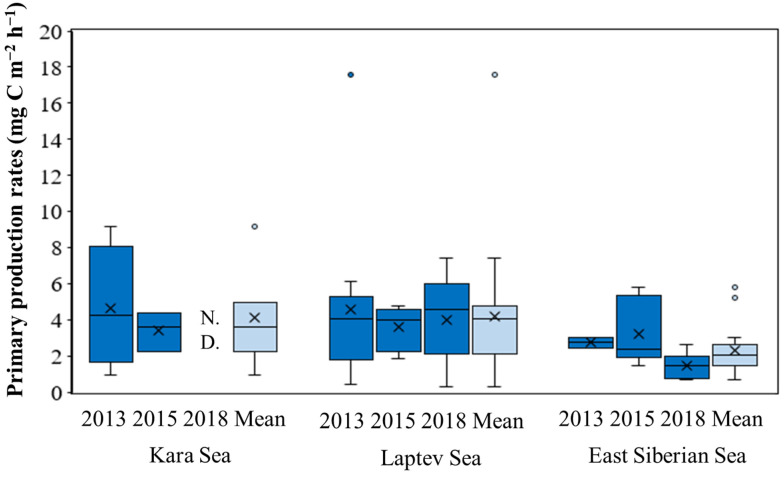
Yearly average primary production rates in the Kara Sea, Laptev Sea, and East Siberian Sea in 2013, 2015, and 2018. N.D.: no data available.

**Figure 9 microorganisms-11-01886-f009:**
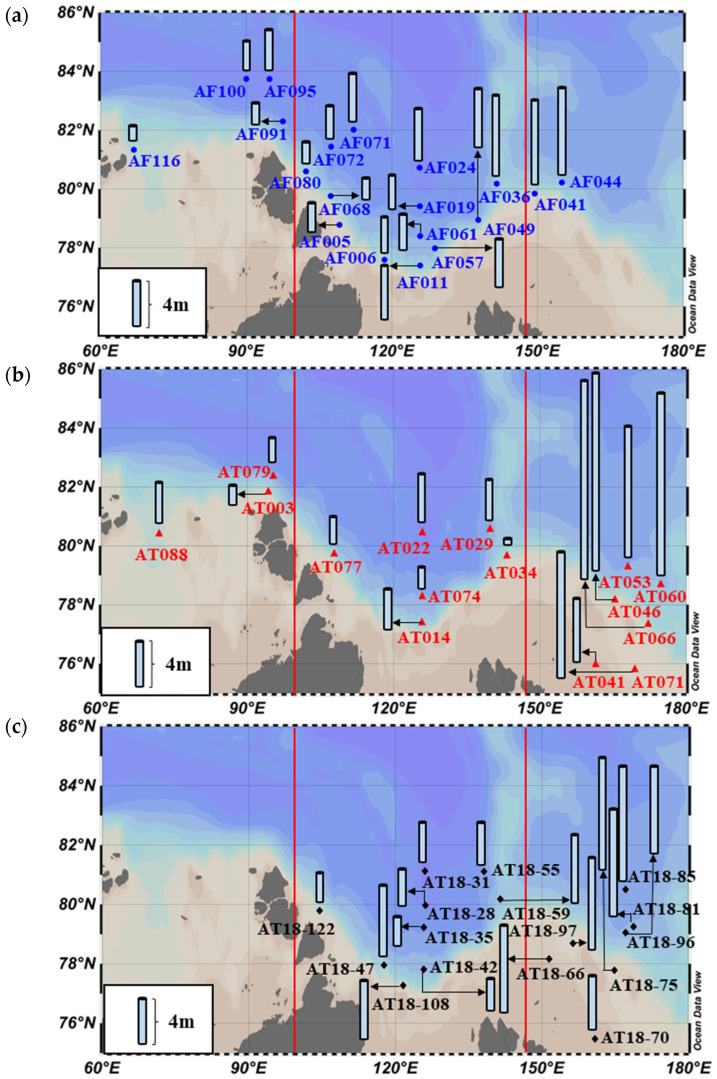
Spatial distribution of the freshwater content during (**a**) 2013, (**b**) 2015, and (**c**) 2018.

**Figure 10 microorganisms-11-01886-f010:**
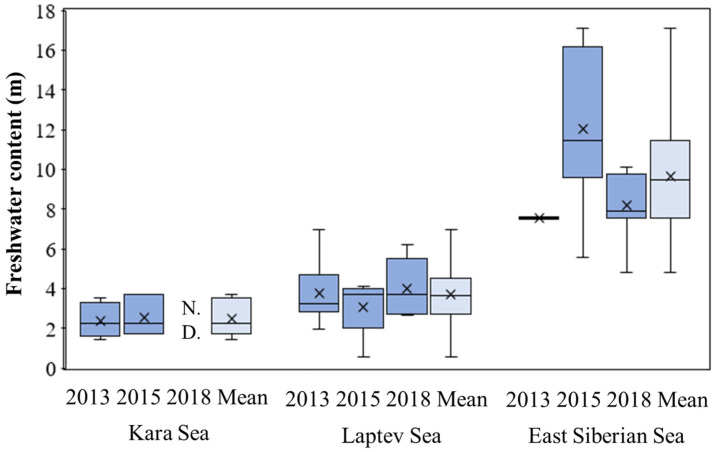
Yearly averaged freshwater contents in the Kara Sea, Laptev Sea, and East Siberian Sea in 2013, 2015, and 2018. N.D.: no data available.

**Figure 11 microorganisms-11-01886-f011:**
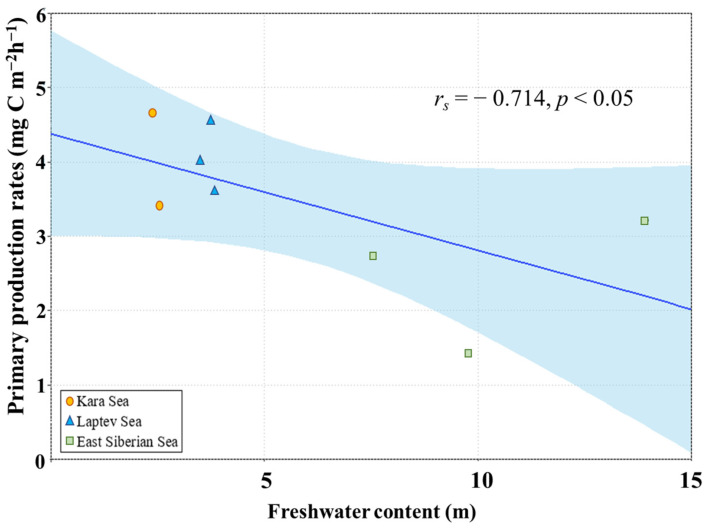
Relationship between freshwater contents and primary production rates.

**Table 1 microorganisms-11-01886-t001:** Locations and physical conditions at the productivity measurement stations in the Kara Sea, Laptev Sea, and East Siberian Sea in 2013.

	Sea	Station Name	Date	Latitude	Longitude	Temperature (°C)	Salinity	Euphotic Depth (m)	Mixed Layer Depth (m)	Stratification Index (kg m^−3^)
2013	Kara Sea	AF091	14 September	82.30	97.55	−1.14 ± 0.70	34.10 ± 0.52	38	31	0.50
		AF095	15 September	83.74	94.79	−1.45 ± 0.32	33.69 ± 0.80	70	19	1.39
		AF100	16 September	83.75	90.01	−1.59 ± 0.06	34.08 ± 0.39	46	17	0.88
		AF116	19 September	81.34	66.87	0.42 ± 0.83	34.28 ± 0.57	46	26	0.82
	Laptev Sea	AF005	25 August	78.78	109.20	−1.16 ± 0.41	33.88 ± 1.13	38	13	2.83
		AF006	26 August	77.59	118.45	−0.91 ± 0.89	33.73 ± 1.06	50	10	2.95
		AF011	27 August	77.40	125.80	−0.91 ± 1.20	33.20 ± 1.55	51	14	4.16
		AF019	28 August	79.42	125.74	−1.43 ± 0.33	33.82 ± 0.71	60	15	1.33
		AF024	29 August	80.72	125.69	−1.47 ± 0.32	33.31 ± 1.17	51	9	2.47
		AF036	1 September	80.18	141.56	−1.58 ± 0.15	32.59 ± 1.96	54	9	4.77
		AF049	5 September	78.95	137.77	−0.91 ± 1.01	33.21 ± 1.71	51	7	4.68
		AF057	5 September	77.98	128.83	−1.26 ± 0.83	33.43 ± 1.17	51	5	3.83
		AF061	6 September	78.40	125.83	−1.31 ± 0.45	33.79 ± 0.87	51	9	2.59
		AF068	10 September	79.76	107.39	−1.32 ± 0.43	34.12 ± 0.59	33	6	1.64
		AF071	11 September	82.02	112.10	−1.44 ± 0.36	33.41 ± 1.00	43	27	1.68
		AF072	12 September	81.44	107.48	−1.39 ± 0.35	33.83 ± 0.80	49	14	1.50
		AF080	13 September	80.60	102.31	−0.14 ± 0.55	34.11 ± 0.71	76	22	1.12
	East Siberian Sea	AF041	2 September	79.85	149.38	−1.49 ± 0.18	32.41 ± 1.47	51	18	3.24
		AF044	3 September	80.22	154.98	−1.57 ± 0.13	32.42 ± 1.25	35	28	2.46

**Table 2 microorganisms-11-01886-t002:** Locations and physical conditions at the productivity measurement stations in the Kara Sea, Laptev Sea, and East Siberian Sea in 2015.

Period	Sea	Station Name	Date	Latitude	Longitude	Temperature (°C)	Salinity	Euphotic Depth (m)	Mixed Layer Depth (m)	Stratification Index (kg m^−3^)
2015	Kara Sea	AT003	28 August	81.84	94.31	−0.77 ± 0.50	34.18 ± 0.50	30	25	0.85
		AT079	24 September	87.42	95.33	−1.60 ± 0.25	34.03 ± 0.81	56	15	1.55
		AT088	26 September	80.42	71.91	−0.07 ± 1.34	33.50 ± 2.77	26	10	7.22
	Laptev Sea	AT014	1 September	77.40	125.80	−0.80 ± 1.35	33.65 ± 1.23	30	8	4.18
		AT022	4 September	80.46	125.89	−1.31 ± 0.65	33.47 ± 1.17	40	18	2.73
		AT029	6 September	80.57	139.78	−1.13 ± 0.84	33.65 ± 0.90	40	10	2.37
		AT034	8 September	79.68	143.24	N.D.	N.D.	28	N.D.	N.D.
		AT074	19 September	78.29	125.87	−0.92 ± 1.40	33.50 ± 1.51	30	21	3.76
		AT077	22 September	79.74	107.82	−1.61 ± 0.11	33.06 ± 1.62	34	11	3.11
	East Siberian Sea	AT041	10 September	75.98	161.43	−1.47 ± 0.09	30.19 ± 1.23	52	18	2.18
		AT046	10 September	78.18	165.37	−1.44 ± 0.22	31.43 ± 1.55	56	8	3.90
		AT053	12 September	79.30	168.03	−1.41 ± 0.22	31.53 ± 1.41	54	10	3.89
		AT060	14 September	78.70	174.77	−1.21 ± 0.33	30.64 ± 2.04	57	15	4.56
		AT066	14 September	77.36	172.19	−1.14 ± 0.33	30.40 ± 2.25	70	17	5.03
		AT071	16 September	75.82	169.46	−1.21 ± 0.58	30.95 ± 2.33	60	13	5.59

N.D.: no data available.

**Table 3 microorganisms-11-01886-t003:** Locations and physical conditions at the productivity measurement stations in the Kara Sea, Laptev Sea, and East Siberian Sea in 2018.

Period	Sea	Station Name	Date	Latitude	Longitude	Temperature (°C)	Salinity	Euphotic Depth (m)	Mixed Layer Depth (m)	Stratification Index (kg m^−3^)
2018	Laptev Sea	AT18-28	31 August	79.97	126.17	−1.61 ± 0.19	33.78 ± 1.10	38	16	2.43
		AT18-31	1 September	81.13	126.08	−1.61 ± 0.12	33.62 ± 1.26	38	11	2.91
		AT18-35	2 September	79.23	125.84	−1.18 ± 0.90	33.92 ± 0.79	48	18	2.12
		AT18-42	3 September	77.81	125.83	−0.73 ± 1.38	33.85 ± 0.95	40	17	2.87
		AT18-47	4 September	77.96	117.69	−0.79 ± 0.78	33.53 ± 1.34	26	5	4.70
		AT18-55	6 September	81.11	138.14	−1.70 ± 0.09	33.59 ± 1.06	36	15	2.20
		AT18-59	6 September	80.18	141.49	−0.88 ± 1.33	32.83 ± 2.04	40	22	4.81
		AT18-108	21 September	77.27	121.62	−0.64 ± 0.78	33.38 ± 1.40	28	6	4.30
		AT18-122	23 September	79.79	104.51	−0.37 ± 0.82	33.87 ± 1.03	34	25	1.75
	East Siberian Sea	AT18-66	8 September	78.17	151.48	0.09 ± 2.14	29.86 ± 1.76	20	21	4.77
		AT18-70	8 September	75.47	160.87	−1.46 ± 0.10	30.26 ± 1.15	46	16	2.30
		AT18-75	9 September	77.78	164.84	−1.39 ± 0.15	31.70 ± 1.87	36	11	4.21
		AT18-81	10 September	79.25	168.78	−1.56 ± 0.09	31.80 ± 1.44	42	25	3.09
		AT18-85	13 September	80.50	167.15	−1.60 ± 0.09	31.75 ± 1.07	40	24	2.60
		AT18-96	17 September	79.05	167.15	−0.76 ± 1.29	32.22 ± 2.14	24	15	5.68
		AT18-97	17 September	78.69	156.39	0.04 ± 1.87	31.59 ± 2.54	22	30	5.62

**Table 4 microorganisms-11-01886-t004:** Comparison between this and previous studies of daily primary production rates in the Kara Sea, the Laptev Sea, and the East Siberian Sea.

Region	Primary Production Rates (mg C m^−2^ d^−1^)	Method	Year	Month	Season	Reference
Kara Sea	55 ± 20	^14^C	2007	September	Summer	[65]
	24 ± 71		2011	September–October	Summer–autumn	
	99 ± 62	^13^C	2013, 2015	August–September	Summer	This study
Laptev Sea	64 ± 19	^14^C	2015	September	Summer	[62]
	23 ± 11		2017	September		
	55 ± 25		2018	August–September		
	100 ± 77	^13^C	2013, 2015, 2018	August–September	Summer	This study
East Siberian Sea	33 ± 15	^13^C	2004–2012	July	Summer	[37]
	93 ± 49			August		
	132 ± 44			September		
	28 ± 13	^14^C	2017	September	Summer	[52]
	56 ± 35	^13^C	2013, 2015, 2018	August–September	Summer	This study

## Data Availability

Not applicable.

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
