# Peer review of "Primary Production in the Kara, Laptev, and East Siberian Seas"

_microorganisms, 2023, doi:10.3390/microorganisms11081886_

Round 1

Reviewer 1 Report

line107, 222, in the paragraphs, the units are in wrong format. should be superscript.

line 232, yearly difference, should be interannual difference. 

 Figure 11, carbon uptake rate should be primary production. In MM and Result, only carbon uptake. However, introduction discussion and title, primary production is used. I recommend use the same word throughout the paper. 

Line 268 yearly  vs.  interannually 

table 4 is missing

Author Response

line107, 222, in the paragraphs, the units are in wrong format. should be superscript.

=>We revised them correctly (Line 131-134). Thank you for pointing them out! For the line 222, we revised the whole section of 3.4 without repeating the numerous numbers as suggested by the other two reviewers.

line 232, yearly difference, should be interannual difference. 

=>We revised it throughout the text including the caption for Fig. 8. 

 Figure 11, carbon uptake rate should be primary production. In MM and Result, only carbon uptake. However, introduction discussion and title, primary production is used. I recommend use the same word throughout the paper. 

=>We replaced carbon uptake rate by primary production throughout the text. 

Line 268 yearly  vs.  interannually 

=>We revised it throughout the text including the caption for Fig. 8. 

table 4 is missing

=>Somehow, Table 4 was missing in the earlier version of our manuscript. We inserted it in our revised manuscript. Very sorry for that! 

Reviewer 2 Report

Review of the manuscript: „Interannual and Regional Variations in Primary Productions in 2 the Kara Sea, the Laptev Sea, and the East Siberian Sea“ by S. Kim et al.

The manuscript describes phytoplankton primary production measurements at three sites in the Arctic Ocean at several years. Such data are important and valuable, especially as the investigated sites are highly dynamic and affected by various effects of global change.

Some critical comments remain:

Light depth measurements:  The authors used a Secchi disk to estimate the depth of 1, 5, 12, 30, 50 and 100% light penetration of surface PAR. These depths will vary according to the underwater light absorption coefficient, the time of the day, waves, and the cloud cover of the sky. Showing some absolute depths values (m) were the different light depths were located would be important and helpful, especially as the data are planned to be published in a more general journal dealing with microorganisms and not an oceanographic journal. I was a bit surprised seeing that for such important measurements, which are used to upscale to large geographical scales and entire years,  a simple Secchi desk was used to measure the most important parameter for primary production, the light. Whereas nothing is wrong with a Secchi disk and it is a valuable instrument, it is still only a white disk with usually 20cm diameter and one observes the depth at which it will not be visible anymore. The measurements with a Secchi disk are therefore the personal observation at which depth he/she does not anymore see the disk when released into the water. It is obvious that this is not fully comparable with a light sensor/meter that is reading direct light levels (µmol photons etc) at a certain depth. There are published equations converting Secchi depths into euphotic depth, however these equations vary a lot, depending on the type of water body where the observations are made. Hence, a minimum requirement would be to give a clear description how the Secchi depth of an individual observer was converted to the so called “light depths” at exactly 1, 5, 12, 30, 50 qnd 100% surface PAR.

Carbon uptake: The authors used a deck incubator for carbon uptake rates estimated in 1l bottles. Bottles were covered with lighting screens to obtain the same light levels such as in the six 1 to 100% light depths measured by a Secchi disk in situ (see comments above). My question is how the authors converted the Secchi depth estimate to estimate the light levels needed for the deck incubation? A light sensor (PAR) would have read the amount of light at a certain depth (µmol photons per m2 per sec) within the water column and it would be easy to adjust the deck incubation light levels to exactly the light level s in situ. Additionally the light level on deck could be checked regularly by such a device. It is not clear to me how this was done with a Secchi disk. To my opinion this is a crucial part as the carbon uptake rates were used to upscale to the entire euphotic water column (production below m2 of water column). Additionally I was wondering how the authors integrated the individual measurements from the deck incubations for estimates of entire euphotic zones. I would have thought that also the mixing intensity and the mixing depth of the water column would be a critical factor (and an average Imix light level)?

-Carbon uptake rates are a central aspect of the manuscript. Hence, I would have expected that some more details(concentrations, etc)about incubation (added concentrations, labeling values etc)   are presented; not only :  “adding reagents of labeled carbon….”

-The authors describe within methods that also labeled nitrate and/or ammonium (?) reagents were added. However I cannot find any results showing nitrogen uptake rates.

Results:

-Fig. 3 shows atomic dissolved (?) nutrient ratios for all sampling stations. Within these graphs areas are indicated by red lines where certain nutrients are limiting primary production. To me it is unclear how the authors estimated nutrient limitation? I cannot find anything about this aspect within methods.  Are estimates of nutrient limitation based on Redfield ratios and is it assumed that uptake of nutrients of phytoplankton follow Redfield ratios?? To my knowledge this must not always be the case, there exist numerous examples of “non Redfield uptake dynamics”. I would have assumed that only rigid nutrient uptake bioassays would allow clear statements about primary production nutrient limitation patterns.

-To my opinion the results part is very difficult to read, over large parts it is more reading like a cruise report with tons of numbers within the main text than as a scientific manuscript submitted to a special issue about “Marine Microorganisms in A Changing Ocean: From Single Species to Community Responses. I think it can be helpful to check whether all numbers given in the main text are necessary or are already included in tables or figure. Additionally, part of it could be probably better presented within tables than within the main text of the manuscript.

Discussion: For my feeling the discussion could be more focused towards the topic of the special issue, to me it is in its current form more suited for an oceanographic journal.

Author Response

 We truly appreciated with your valuable comments on the earlier version of our manuscript. We have revised the manuscript incorporating virtually all provided comments as attached file. Especially, we tried to extend our methods for light depth and carbon uptake measurements to make it clearly. Our detail responses corresponding to your comments in our attached file.

Thanks!

Reviewer 3 Report

This is an interesting paper and an important one.  It must be published, but it cannot in its present state.  Part of this is the manner the manuscript is presented, which is the responsibility of the journal.  Beside that one cannot improve the language because of the lack of space the editor did not look at the figures.  He/she should have seen that they are not readable because most of them are too small (Fig. 1, 2, 3 4, 7, 9).  Also, the editor forgot Table 4.  Tables 1 to 3 take more space than the text.  Why is this so?

The title should be shortened.  It is primary production (singular) and please write Kara, Laptev and East Siberian Seas.

You need to improve the English to remove odd things.  I am sorry to mention this (but English is also not my mother tongue).

I found about 20 relevant English primary production publications from Siberia by Demidov, but you cite only two.  That is not good enough. More is known form Siberia. There are modeling results from Slagstad that you may look into.

You have valuable data from 3 years but you interpret them as if the timing of the bloom was the same each year.  This is probably not the case.  And you measured in early fall, but most of the primary production (and in particular new production) takes place in summer.  You have no idea about this but dare to provide annual estimates. Some of your estimates must be too low to run an entire ecosystem.  Compare with Arrigo & Co and Slagstad.  

The insert in Fig.1 tells nothing!  Where are we?  Only specialist can tell.

Your manuscript would have benefitted if you had give it to experiences scientist such as Ardyna.

You show a large number of rates in the results and it is difficult to follow.  Can you place the results into small tables?

The low production in the East Siberian Sea may be the effect of the freshening of the Beaufort Gyre.  Please check and discuss.

What is the difference between Carbon uptake rates and primary production?

The editor was not handling 15N and 13C right.  This must be corrected.

I think that you have not done your job good enough and therefore this paper needs to be fully revised.

Why is the order of authors different in the files and in the manuscript?

The English is pretty good but needs to be improved.

"significant levels of dynamic activity in terms of synthesis and processing organs matter". What do you mean?

There are more "strange" formulations.

Whitledge and Stockwell could have read this.....

Author Response

 We truly appreciated with your valuable comments on the earlier version of our manuscript. We have revised the manuscript incorporating virtually all provided comments. Especially, we revised our manuscript to make it readable. Our detail responses corresponding to your comments in our attached file. Thank you!

Round 2

Reviewer 2 Report

The authors followed in detail the suggestions of the referees. To my opinion they did this in an convincing way.  Some critical aspects  regarding methods can not be improved as either necessary infrastructure or measurements were not available, however these aspects are now clarified within the revised manuscript.

Author Response

We truly appreciated with your valuable comments on the earlier version of our manuscript. We revised our manuscript to clarify some methodological aspects. Thanks!